# Overview of the Complex Relationship between Epigenetics Markers, CTG Repeat Instability and Symptoms in Myotonic Dystrophy Type 1

**DOI:** 10.3390/ijms23073477

**Published:** 2022-03-23

**Authors:** Laure de Pontual, Stéphanie Tomé

**Affiliations:** Centre de Recherche en Myologie, Inserm, Institut de Myologie, Sorbonne Université, F-75013 Paris, France; l.depontual@institut-myologie.org

**Keywords:** myotonic dystrophy type 1 (DM1), DM1 phenotype, CTG repeat expansion, DNA methylation and chromatin, DM1 prediction and therapeutic targets

## Abstract

Among the trinucleotide repeat disorders, myotonic dystrophy type 1 (DM1) is one of the most complex neuromuscular diseases caused by an unstable CTG repeat expansion in the *DMPK* gene. DM1 patients exhibit high variability in the dynamics of CTG repeat instability and in the manifestations and progression of the disease. The largest expanded alleles are generally associated with the earliest and most severe clinical form. However, CTG repeat length alone is not sufficient to predict disease severity and progression, suggesting the involvement of other factors. Several data support the role of epigenetic alterations in clinical and genetic variability. By highlighting epigenetic alterations in DM1, this review provides a new avenue on how these changes can serve as biomarkers to predict clinical features and the mutation behavior.

## 1. Introduction

Neuromuscular disorders (NMD) include diseases affecting the peripheral nervous system and the skeletal muscles caused by either gene mutations or immune system abnormalities [1]. Among NMDs, trinucleotide repeat (TNR) disorders caused by an abnormal triplet repeat sequence expansion, such as X-linked spinal and bulbar muscular atrophy (SBMA), oculopharyngeal muscular dystrophy (OPMD) and myotonic dystrophy type 1 (DM1), have been identified [2]. DM1 is the most common adult muscular dystrophy with a genetic prevalence of 1 in 2100 in the New York State population [3]. DM1 is an autosomal dominant and multisystemic disease with variable symptoms including muscle weakness, myotonia, cataracts, diabetes, cardiac conduction defects, respiratory distress and cognitive impairments [4]. The complex and highly variable symptoms of DM1 are classified into five clinical forms (congenital, infantile, juvenile, adult and late onset) based on symptom severity and age of onset [4]. Some symptoms, such as muscle weakness and cognitive impairments, are commonly seen in congenital forms, while others, such as cataracts and diabetes, are more specific to adult or late-onset forms (Figure 1) [4]. Within the five forms of DM1, the type and severity of symptoms vary widely between DM1 families but also between members of the same family.

DM1 is caused by an unstable CTG repeat expansion in the 3′-untranslated region (UTR) of the *Dystrophy Myotonic Protein Kinase (DMPK)* gene on chromosome 19q13.3 [5,6,7,8,9]. In healthy individuals, the repeat length is polymorphic and usually ranges from 5 to 37 CTG whereas the number of repeats is greater than 50 CTG and can reach up to several thousand in DM1 patients [10]. The pathogenic CTG repeat is highly unstable and increases in length with each generation (intergenerational instability) and in tissues (somatic instability) [11,12,13]. The expansion-biased intergenerational instability of the CTG repeat provides a molecular explanation for the phenomenon of anticipation that is defined by an increase of the disease severity and a decreasing age of onset over successive generations [13,14,15]. Larger repeats are generally associated with a more severe phenotype and an earlier age of onset (congenital form), whereas smaller repeats are commonly found in individuals with a late-onset form [4,14]. However, CTG repeat size alone cannot explain disease severity and progression, strongly suggesting that other factors play a role in clinical manifestations, contributing to phenotype variability [4,13]. The gender of DM1 patients has a significant impact on the disease phenotype where males more frequently exhibit myotonia, muscle weakness, cardiac and respiratory disabilities as well as facial dysmorphism and cognitive impairments. In contrast, cataracts, digestive symptoms, dysphagia and thyroid disorder are more common in DM1 females [16,17]. The sex of the transmitting parent also affects the dynamics of intergenerational instability and thus the severity of the disease. DM1 congenital forms are usually observed in maternal transmissions. This is explained by the size of the CTG repeat expansion but also by the methylation patterns surrounding the repeats [18,19].

The degree of somatic instability observed in DM1 patients is highly biased towards expansion, age-dependent and tissue-specific [13,20,21,22,23]. Somatic instability begins between 13 and 16 weeks of gestation and pursued throughout patient life [12,24,25]. In fetal tissues, the largest expansions were found in the heart and muscle and the smallest one in the liver, whereas in adult DM1 tissues, the largest expansions were identified in the heart and cerebral cortex and the smallest in cerebellum [12,26]. Furthermore, somatic instability levels in blood have been shown to correlate with age of onset and thus with disease severity [23,27,28]. Many factors have been shown to modify repeat instability including CTG repeat size and CNG repeat interruptions [27,28,29]. Indeed, CNG interruptions have been reported in the expanded CTG tract of 4–9% of DM1 patients and associated with stabilization of CTG repeat length in blood [27,29,30,31,32,33,34,35,36]. The presence of an interrupted CTG tract is associated with a milder phenotype and late disease onset [29,32,35,37,38]. *MutS Homolog 2* (*MSH2)* and *3* (*MSH3)*, two DNA mismatch repair genes required to maintain genome integrity, also play a major role in the formation of CTG repeat expansions in DM1 mice [13,39,40,41], In DM1 patients, single-nucleotide polymorphisms in *MSH3* have been associated with a decreased level of somatic mosaicism and a decreased age of onset [42,43]. The level of somatic instability could also be explained by tissue-specific DNA methylation in DM1. Methylation profiles near CTG repeats in DM1 patients have been shown to be highly variable both between identical tissues of patients and within tissues of the same patient [12]. CTG repeat instability and the DM1 phenotype are highly dependent on the length and purity of repeat expansions, DNA replication and repair proteins but also epigenetic modifications (methylation) and chromatin structure (CTCF) [12,29,43,44,45,46,47,48].

The CTG repeat expansion affects the expression of *DMPK*, as well as *Sine Oculis Homeobox Homolog 5* (*SIX5*) and possibly *Dystrophia Myotonica WD Repeat-Containing* (*DMWD)*, located respectively, downstream and upstream of the *DMPK* gene (Figure 2a) [49,50,51,52,53]. In DM1, downregulation of these genes by epigenetic changes, such as hypermethylation may participate in some clinical features including cataracts, muscle defects or cardiac conduction abnormalities as observed in *DMPK*, *SIX5* or *DMWD* knockdown mouse models [54,55,56,57,58,59]. However, loss-of-function of *DMPK* and its flanking genes cannot explain the majority of clinical manifestations observed in DM1 patients, suggesting a role of the mutation in *trans*. CTG expansions in *DMPK* transcripts induce accumulation and sequestration of mutated *DMPK* mRNAs in cell nuclei, forming toxic ribonuclear inclusions called foci (Figure 2b) [60,61]. Retention of mutated *DMPK* transcripts alters the levels and localization of important splicing regulators such as Muscleblind-Like Protein 1 (MBNL1) and CUGBP Elav-Like Family Member 1 (CELF) proteins, inducing splicing defect of several genes including *Chloride Voltage-Gated Channel 1* (*CLCN1*), *Insulin Receptor* (*INSR*) or *Bridging Integrator 1* (*BIN1*) responsible for cardiac conduction defect, diabetes and muscle weakness, respectively (Figure 2c) [60,61,62,63].

In DM1, the genotype-phenotype association remains unclear due to high genetic and clinical variability in patients, making prognosis, implementation of personalized treatments and genetic counseling extremely difficult. In this review, we will address the effects of epigenetic alterations on the clinic and genetics of DM1 and discuss how epigenetic modifications could serve as predictive biomarkers to improve diagnosis, prognosis and therapies in this disease (Figure 2d).

## 2. Epigenetic Modifications and DM1 Clinical Features

In DM1, CTG repeat size gives an indication of the likely disease age of onset and on the clinical form. However, the considerable overlap in CTG repeat size between different clinical forms indicates that size is not sufficient to predict symptoms in DM1 patients and therefore suggests the contribution of other factors such as DNA methylation [4].

### 2.1. CDM1 Form Associated with Hypermethylation of the DM1 Locus and Large CTG Repeat Expansions

The congenital DM1 (CDM1) form is usually associated with a large CTG repeat size (>1000) but can also be associated with shorter CTG repeat tracts, which often leads to difficult genetic counseling in DM1 patients [4]. A common hypothesis to explain the overlap in CTG repeat size between congenital and non-congenital DM1 is the existence of CDM1-specific epigenetic marks. Despite the difficulties of analyzing epigenetic marks in DM1 and the variability observed between studies resulting from different methods, samples (tissue types and donors) and imprecise description of clinical features and CTG repeat size, several studies have shown that hypermethylation upstream of repeat expansion is, in most cases, associated with earlier onset and more severe manifestations of the disease [12,18,23,64,65]. By methylation-sensitive endonuclease digestion of leucocytes, Steinbach et al. identified hypermethylation of various CpG sites in the 1.8 Kb upstream of CTG repeats in patients with a congenital form whereas only constitutive normal methylation was found in patients with other forms of DM1, suggesting a role of methylation in the development of early DM1 forms and in particular of CDM1 form [65]. Spits et al. analyzed more precisely 8 out of the 18 CpG sites in the 152 nucleotides upstream of the CTG repeat site in 22 DM1 patients with different DM1 clinical status including two CDM1 using methylation-sensitive endonuclease digestion [66]. They found that upstream methylation was not limited to the largest CTG repeat expansion and congenital form in their DM1 cohort which is in contradiction with Steinbach et al. [65,66]. These differences can be explained by the cohort sizes in which the clinical and genetic status of each patient was not fully characterized.

To explore the potential association between hypermethylation of the DM1 locus and the CDM1 form and its potential use as a biomarker, more recent studies used more precise quantitative methods in new DM1 samples. By methylation sensitive high-resolution melting (MS-HRM) upstream and downstream of the repeat expansion, Santoro et al. showed that hypermethylation occurs only upstream of the repeat and found a strong association between upstream methylation and larger repeat sizes as well as an earlier age at onset and a maternally inherited mutated allele in blood samples from a cohort of 66 DM1 patients [64]. However, disease severity, as measured by Muscular Impairment Rating Scale (MIRS), was not associated with methylation status around the CTG repeats [64]. More recently, Barbé et al. analyzed the methylation in the flanking region of the CTG repeat expansion in a large cohort of DM1 patients, including 20 CDM1 patients [18]. In this study, 19/20 CDM1 patients carrying from 1110 to 4700 repeats showed hypermethylation upstream of the CTG repeats but also downstream of the repeats [18]. Using pyrosequencing-based methylation analysis (PMA), Morales et al. showed that immediate upstream and downstream regions flanking the repeat were also hypermethylated in two CDM1 patients and one pediatric DM1 case while no hypermethylation was found in the remaining 40 patients (34 adult/late onset DM1 patients and 6 pediatric cases) [23]. Later, the same group extended their work by conducting a similar but larger study of 225 DM1 patients from 62 families [19]. Of the 22 CDM1 patients, 19 (86%) harbored hypermethylated upstream and downstream regions. Of the 30 pediatric cases in the study, 7 (23%) were hypermethylated upstream and 3 (10%) downstream, whereas only 5 and 3 out of 140 DM1 patients with adult forms had hypermethylation respectively upstream and downstream of the repeats [19]. Finally, no late-onset DM1 patients or asymptomatic cases showed hypermethylation [19]. Hypermethylation was only observed in patients with expansion of more than 518 CTG repeats (Estimated progenitor allele length) regardless of clinical forms [19]. Interestingly, Barbé et al. also showed that DM1 patients with more than 1000 CTG repeats and less severe forms of DM1 show mainly no hypermethylation in the flanking sequence of the CTG repeat, suggesting that methylation is not fully associated with repeat size [18]. Lopez-Castel et al. used bisulfite sequencing to determine the methylation profile at each CpG site around the CTG repeat expansion in various human tissues with more than 2000 CTG repeat from CDM1 and DM1 patients [12]. The authors showed that the methylation status upstream of the repeat is dependent on tissue type, independently of repeat size [12].

The studies described above have shown that hypermethylation of the DM1 locus is preferentially associated with large CTG repeat expansion (>500 CTG) and the CDM1 form [18,19,23,64]. The hypermethylated DM1 locus in patients with a large CTG repeat expansion could be used as a biomarker for the CDM1 form. However, current knowledge does not support the use of methylation alone as biomarker due to the description of non-CDM1 patients with hypermethylation of the DM1 locus and some CDM1 patients with no hypermethylation [12,18,19,64,66]. In the future, it would be important to simultaneously study DM1 locus methylation status and repeat size in a large DM1 cohort using new state-of-the-art methods such as long-read sequencing developed by Oxford Nanopore and Pacific Biosciences in order to precisely characterize DM1 locus methylation status on a base-by-base and to be able to use it as biomarker if possible [67,68,69].

### 2.2. DM1 Symptoms Associated with Specific Epigenetic Patterns

DM1 is a multisystemic disease that affects all organs with high clinical variability resulting in great difficulty to predict the different symptoms and their progression. Two recent studies quantified methylation status of 57 CpG sites at the *DMPK* locus in blood of 90 and 115 DM1 patients with adult or late-onset forms using the bisulfite pyrosequencing method [44,48]. Légaré et al. showed that DNA methylation at two CpG sites downstream of the CTG repeat expansion correlates with respiratory failure and muscle weakness, independently of the CTG repeat length [48]. Similarly, a longitudinal study conducted over 9 years of follow-up in blood of 115 DM1 patients with adult form suggested that DNA methylation levels downstream of the CTG repeat were associated with changes in cognitive features [44]. In these two studies, the association between methylation at the *DMPK* locus and muscle, respiratory or cognitive phenotypes is highly dependent on methylation pattern of specific CpG sites. To complete these studies, a quantitative methylation method at each CpG site of the *DMPK* locus coupled with a detailed analysis of patient phenotypes in a large DM1 cohort is crucial to validate DNA methylation as biomarker for specific symptoms.

In addition to DNA methylation, histone changes and non-coding RNA-associated gene silencing, such as circular RNA, are identified as two other epigenetic mechanisms playing a role in molecular and cellular processes. Circular RNAs (circRNAs), highly stable single-stranded RNA, form covalently closed loops by joining the 3′ and 5′ ends of conventional RNA molecules [70]. CircRNAs have already been identified as regulators of transcription, alternative splicing and chromatin interactions [71]. In quadriceps femoris and anterior tibialis of DM1 patients, an overall increased level of circRNAs was observed compared to control individuals [72]. CircRNAs upregulation, both global or of specific circRNA, has been negatively associated with muscle strength in DM1 patients biopsies [72,73]. Moreover, Voellenkle et al. identified 4 DM1 upregulated circRNA in muscles biopsies and demonstrated that their expression level was sufficient to discriminate DM1 patients and control individuals [73]. In contrast to muscle, circular RNA levels are similar in DM1 and control samples in peripheral-blood-mononuclear cells and in plasma samples, suggesting that circRNAs upregulation is tissue-specific [73]. To conclude, the exact role of circRNAs on DM1 pathogenesis remains unknown, but recent studies have clearly shown that circRNAs levels in muscle could serve as a biomarker of the disease, as well as a phenotypic biomarker to estimate the level of muscle defects in DM1 patients.

## 3. Sex, CTG Repeats and Epigenetics

### 3.1. DNA Methylation at the DMPK Locus May Contribute to Sex Differences in DM1 Symptoms

In a large French DM1 registry (1409 DM1 patients), it has been shown that the clinical profile in men and women is different in the DM1 population [16]. Men have a higher risk of developing symptoms such as severe myotonia, muscle weakness, respiratory impairment, cardiac conduction defect and facial dysmorphism than women [16]. On the other hand, female patients suffer more often from cataracts, dysphagia, thyroid disorder and gastrointestinal problems [16]. More recently, a higher prevalence of gastrointestinal symptoms in female patients and a higher risk of cardiac conduction abnormalities in males were also observed in two independent DM1 cohorts with 103 and 61 DM1 patients, respectively [17,74].

The clinical differences observed in the DM1 population according to the sex of patients could be explained by different factors among which epigenetic processes such as methylation. In unaffected individuals (96 males and 96 females), several loci, including loci downstream of the *DMPK* region, were found to be slightly hypermethylated in the blood of men [75]. Recently, it has been shown that DNA methylation levels and gene expression in myotubes, myoblasts and skeletal muscle biopsies differ between men and women [75]. Interestingly, one of these genes with sex-differentiated expression, *KDM6A*, is a histone demethylase known to regulate certain genes during myogenesis [76,77]. These results suggest that DM1 patients may be more sensitive to certain symptoms depending on their sex-linked methylation pattern. The epigenetic and transcriptional differences observed between males and females could contribute to sex-dependent differences in DM1 muscle phenotype, in particular to the increased muscle disability observed in male DM1 patients [16].

Furthermore, most male DM1 patients have reduced fertility and hypogonadism that could be explained by epigenetic and expression changes of genes involved in sperm quality [14,78,79,80]. *RSPH6A,* a testis-specific gene located 13 kb upstream of *DMPK*, is hypomethylated in sperm from general population and potentially hypermethylated in DM1 sperm that show hypermethylation downstream of the CTG repeat expansion (Figure 2a) [45,81]. Ergoli et al. showed that *RSPH6A* expression is decreased in DM1 sperm compared to controls [79]. The exact role of *RSPH6* is currently unknown, however, Paudel et al. recently showed that RSPH6 is hyperphosphorylated during sperm capacitation in mice, a maturation step that allows spermatozoids to penetrate and fertilize eggs [82]. Dysregulation of *RSPH6* could therefore impair mature sperm production [83,84]. Although there is no study showing a direct association between DNA methylation status and *RSPH6* gene expression in DM1 patients, methylation changes might explain the decreased fertility observed in some DM1 men [79].

### 3.2. DNA Methylation Alters the Dynamics of CTG Repeat Instability- Role in the DM1 Phenotype?

DM1 patients showed expansion-biased intergenrational instability, providing a molecular explanation for the anticipation phenomenon defined by increased severity and earlier age of onset over generations [14]. The dynamics of instability depends on the sex and length of the CTG repeats of the transmitting parents. Larger expansions are often maternally transmitted, which explain the maternal transmission bias of CDM1. Recent studies have described that upstream methylation at the DM1 locus was mainly found in patients carrying a maternally inherited expanded allele (>500 CTG) in which epigenetics modification could be responsible for the discrepancy in CDM1 transmission risk between maternal and paternal transmissions [18,19,64,85].

Somatic mosaicism also contributes to the progressive nature of the disease in DM1 [27,28]. DM1 patients with the most rapid increase in triplet size over time in the blood develop earlier and more severe symptoms [23,27,28]. Furthermore, an increase in somatic mosaicism in the blood of the same individual over time is also associated with a change in the age of onset [23]. DNA methylation may contribute to disease development and progression by modulating the dynamics of CTG repeat instability in DM1 patient tissues, either through methylation changes at the DM1 locus or elsewhere in the genome by altering the activity of genes involved in CTG repeat instability. Indeed, genome-wide demethylation using the DNA methyltransferase (DNMT) inhibitor 5-aza-deoxycytidine destabilizes the CTG repeat toward expansions in DM1 human primary fibroblasts [86]. Furthermore, increased DNA methylation downstream of the repeat was associated with lower somatic mosaicism (estimated as the difference between modal allele length and progenitor allele length) in blood samples from 90 DM1 patients analyzed by pyrosequencing [48]. In summary, DNA methylation might be involved in DM1 phenotypic variability through the destabilization of CTG repeat via an unknown direct or indirect mechanism.

### 3.3. DM1 Locus Methylation Deregulated in Patients with an Interrupted Expanded Allele and Less Severe Symptoms

The majority of DM1 patients inherit a pure expansion of CTG repeats. However, 4–9% of the DM1 population carries CTG repeat interruptions, which differ in number and/or type across generations and tissues [27,29,30,31,32,33,34,35,36,37,38,87,88,89]. Repeat interruptions are associated with stabilizations and/or contractions of the CTG repeat expansion and generally milder symptoms [31,32,36,37,64,87,88,89]. The mechanisms by which interrupted expanded alleles stabilize the repeat and then decrease the severity of DM1 symptoms in patients remain poorly understood. DNA methylation in the vicinity of the CTG repeats has been studied in DM1 patients carrying an interrupted expanded allele compared to DM1 patient with a pure CTG repeat tract. By bisulfite sequencing, no difference in CpG methylation pattern between two patients with 150 and 300 interrupted CTG repeats and matched DM1 patients with a pure repeat was found [36]. However, Santoro et al. showed a partially reversed methylation profile compared to pure repeat: increased CpG methylation downstream but no upstream CpG methylation in DM1 patients carrying 400–900 interrupted CTG repeats [64]. Similarly, increased downstream methylation have also been found in other studies in patients carrying interrupted repeats [44,48,85]. Taken together, these observations suggest that interruptions could cause a change in the CpG methylation profile. However, it is difficult to conclude whether this change in methylation profile plays a role in the lower level of somatic instability of interrupted CTG repeats and milder DM1 symptoms or whether it is only a consequence.

## 4. Epigenetics, Gene Expression and Chromatin at the DM1 Locus

Several studies have shown that abnormal CTG repeat expansion promotes DNA hypermethylation and heterochromatin formation at the DM1 locus, making this locus less accessible to transcription factors and eventually causing downregulation of *DMPK* and its neighboring genes, *SIX 5* and *DMWD* [47,49,50,51,52,53,90,91,92]. Although decreased gene expression at the DM1 locus is not sufficient to explain the complex DM1 phenotype, downregulation of *DMPK*, *SIX5* and *DMWD* genes by epigenic changes in DM1 could participate in some clinical features such as DM1 cataracts, muscle defects or cardiac conduction defects. Indeed, generation of mouse models has shown that decreased *DMPK* expression induces late myopathy and mild cardiac conduction defects [54]. However, recently, the generation of a novel *DMPK* knockout mouse model did not reveal any significant defects in cardiac and skeletal muscles [93]. Such a difference between studies could be explained by different parameters such as a different genetic background between models or a difference in model characterization techniques. Earliest studies performed heavy anesthesia on mice while it is now known to disrupt the ECG [94,95]. Furthermore, selection cassettes, used to generate some knockout *DMPK* mouse models, are known to possibly interfere with the expression of flanking genes [96]. Haploinsufficiency of the *SIX5* gene induces cataracts with incomplete penetrance as well as mild cardiac conduction defects [55,57,58]. In a *DMWD*+/− mice, a dramatic reduction of myofiber cross-sectional area was observed, suggesting a potential role of *DMWD* dysregulation in the DM1 muscular phenotype [59].

As suggested above, hypermethylation at the DM1 locus has often been described in DM1 patients with a large CTG repeat expansion despite high variability between patients [12,18,19,23,64,65,66]. CpG methylation, which is mainly observed in the most severe DM1 forms, prevents the binding of CTCF (CCCTC binding factor) on its binding-sites which are located on both sides of the CTG repeats [47]. The CTCF protein function as transcriptional activator or repressor but also as an insulator protein [46,47,97,98]. In DM1, the absence of CTCF binding due to CpG methylation, induces the loss of its insulator activity, eventually allowing the propagation of heterochromatin and thus the downregulation of *DMPK* and *SIX5* genes for more details, see the review [99]. Recently, Yanovsky-Dagan et al. identified a CpG-rich region, 650 bp upstream of the repeat, as an enhancer of *SIX5* [100]. Interestingly, hypermethylation of this region was associated with reduced expression of *SIX5* from the expanded allele in hESC lines, patient-derived iPSCs and derived cardiomyocytes [100]. Thus, the hypermethylation of the DM1 locus could contribute to the disease pathogenesis by causing haploinsufficiency of *SIX5* as illustrated in a *SIX5* heterozygous mice model presenting mild cardiac conduction abnormalities [58].

The presence of CTG repeat affects not only the methylation of the DM1 locus but also the chromatin status of this region. In the 1990s, early studies found a reduction of DNAse I accessibility at the DM1 locus and an increase in nucleosome density in different models such as mice and bacteria, suggesting an alteration of the chromatin structure in DM1 towards heterochromatinization [51,91,92,101]. Our laboratory has created a DM1 transgenic mice carrying 45 kb human genomic DNA including *DMWD*, *DMPK* with over 300 CTG repeats and *SIX5*, cloned from a DM1 patient [102,103,104]. In this DM1 model, Brouwer et al. found increased H3K9 methylation in the hearts of mice carrying >1000 CTG repeats, confirming heterochromatinization of the DM1 locus with large repeats [90]. DM1 cardiomyocytes derived from induced pluripotent stem cells (iPSCs) also showed heterochromatinization around the expanded CTG repeats compared to control DM1 [105]. More recently, it has been shown that CTG repeat expansion is not sufficient to alter 3D chromatin conformation in lymphoblastoid cell lines with more than 1000 CTG repeats, despite changes in CTCF occupancy and DNA methylation levels [106]. In summary, the chromatin structure and methylation of the DM1 locus could participate in the pathogenesis of DM1 by reducing expression of genes at the DM1 locus, but, to date, no study directly prove this hypothesis.

## 5. Epigenetic and Therapy for DM1

### 5.1. Modulation of Somatic Mosaicism by Global Demethylation in DM1 Models

Somatic mosaicism is associated with DM1 disease progression in patients [23,27,28]. As described in the paragraphs above, a recent study showed that DM1 patients with hypermethylation downstream of the CTG repeat expansion tend to have lower somatic mosaicism and thus less severe symptoms [48]. Alteration of methylation at CpG sites as a potential therapeutic target was explored in the early 2000′s in various models. Global hypomethylation with methyltransferase inhibitors such as 5-azacytidine, 5-aza-deoxycytidine (5-aza-CdR) and hydralazine induces a reduction in the expansion rate or a significant increase in the contraction rate in treated DM1 murine kidneys cells (160 CTG) or modified CHO cells (95 CTG.CAG), respectively [86,107]. However, 5-aza-CdR destabilizes trinucleotide repeats towards expansions in two primary human DM1 fibroblasts carrying either 50–80 or 500 CTG repeats (estimated size in blood) [86]. Global hypomethylation alters somatic mosaicism in a model-dependent manner through direct or indirect mechanisms. This process might affect methylation pattern around the repeats or within other genes involved in CTG repeat instability mechanisms [40,41,42,43]. Today, it is difficult to consider broad-range methyltransferase inhibitors as possible treatments for DM1 because they may induce unknown adverse effects. However, exploring the underlying mechanisms by which CpG methylation changes modulate CTG somatic instability is an interesting therapeutic prospect to reduce somatic instability in patients and thus improve their quality of life. To this end, it would be worth targeting the methylated CpG sites around CTG repeats to study the effect of this specific methylation on somatic instability. In the near future, targeted DNA cleavage technologies should allow to precisely edit site-specific DNA methylation, however, for now, off-target effects remain a major challenge [108,109,110].

### 5.2. Modulation of CTG Somatic Instability by Inhibition of Histone Deacetylase

The function of histone deacetylases (HDACs) is to remove acetyl groups from histones and other proteins, which has functional consequences on chromatin structure and gene expression profiles. Several studies have shown the efficacy of HDAC inhibitors in trinucleotide expansion diseases such as polyglutamine diseases [111,112,113]. More recently, it was shown that inhibition of HDAC3 or HDAC5 with small molecules or siRNAs induced a significant reduction in expansion frequency in modified human astrocyte cell lines containing CTG repeats [114,115]. In the same model, Williams et al. demonstrated that RGFP966, a selective HDAC3 inhibitor, affects repeat instability by deacetylating MSH3 protein, a protein involved in the formation of repeat expansions in DM1 mouse models [40,116,117]. Inhibition of HDACs could contribute to attenuate DM1 disease worsening by modulating repeat instability or another pathway that needs to be identified. However, further studies in murine models are needed to ensure that the use of HDAC inhibitors is truly beneficial to the patient without inducing deleterious side effects.

### 5.3. Modulation of MBNL1 Transcription by a Methyl Sensitive Enhancer

In DM1, toxic *DMPK* RNAs with CUG expansion induce a sequestration and inhibition of the RNA-binding protein MBNL1 that contributes to aberrant splicing providing an explanation for many of the symptoms observed in DM1 patients [62,118,119]. Previous studies have shown that increasing MBNL1 proteins or mRNA levels by a minimum 1.5-fold were sufficient to partially reverse aberrant splicing and ameliorate the phenotype of DM1 mice [120,121]. Recently, it was shown that demethylation of MeR2, a transcriptional enhancer located in the intron 1 of *MBNL1*, promotes *Mbnl1* transcription in C2C12 myoblasts as well as in mouse skeletal muscles [120,122]. Thus, acting on the epigenetic processes to increase *MBNL1* transcription and restore free MBNL1 protein level, may be considered as a potential new therapeutic target to restore normal splicing and reduce DM1 symptoms.

## 6. Discussion and Conclusions

DNA methylation and chromatin structure are two altered parameters in DM1 with an overrepresentation of studies analyzing the methylation of CpG sites around CTG repeats [12,18,19,23,47,51,64,65,91,92]. Some data remain controversial and result from the variability observed between methods, regions analyzed and genetic and clinical description of patients. Despite this observed heterogeneity between studies, most studies show that the region upstream of CTG repeats is generally hypermethylated in CDM1 patients compared to unaffected individuals and patients with less severe forms [12,18,19,64,65,85]. To date, the use of methylation alone as a biomarker seems difficult without estimation of the size of the inherited allele by a state-of-the-art method in patients. However, large CTG expansions associated with DNA hypermethylation around the expanded CTG repeat are strong indicators in favor of CDM1. These two indicators would be a powerful tool to predict the most severe forms of DM1 and improve prenatal diagnosis and genetic counseling. Moreover, two recent papers have associated hypermethylation of CpG sites at the DM1 locus with symptoms affecting respiratory, muscular and cognitive functions. These data suggest that accurate mapping of CpG methylation would be necessary to improve prognosis and thus propose earlier treatments to patients to limit disease worsening [44,48]. Interestingly, DNA hydroxymethylation has not been studied in DM1 patients. Recently, aberrant DNA hydroxymethylation changes have been observed in cancers and neurological diseases such as Alzheimer’s disease suggesting an important role of the epigenomic mark hydroxymethylcytosine (5hmC) in tumorigenesis and the neuronal system [123]. 5hmC plays a key role in different pathways such as initiation of demethylation and gene expression regulation in mammals [124,125,126,127]. Analysis of DNA hydroxymethylation at the DM1 locus may provide a new insight into the function of epigenetic modifications in the dynamics of repeat instability but also in the pathogenesis of DM1. Although promising, the use of methylation as biomarker to refine prognosis and improve treatments must be taken with caution as various external and internal factors such as the environment or the patient’s age at the time of sampling are known to affect methylation levels [11,23,128,129,130,131]. In addition, epigenetic biomarkers are usually analyzed in the whole blood of patients. However, epigenetic alterations are known to be tissue-dependent. Therefore, it may be difficult to associate epigenetic markers in blood with the multisystemic DM1 symptoms and the dynamics of CTG repeat instability, which vary from tissue to tissue.

Chromatin structure and gene expression modulated by epigenetic processes are less studied in DM1. It is not yet known whether these parameters could be used as biomarkers in DM1. However, they remain interesting potential therapeutic targets to alleviate DM1 symptoms by epigenetic regulation of activator or repressor of dysfunctional DM1 proteins (MBNL1) or reduction of somatic instability (HDAC inhibitors) [114,115,117,120,122].

Analyses of epigenetic modifications are not straightforward and require homogenization of data through better characterization of the DM1 mutation, DM1 clinics and methylation status at the DM1 locus. The emergence of long-read sequencing technologies in DM1 offers new perspectives to characterize DM1 expansion but also methylation (DNA methylation and hydroxymethylation) at the DM1 locus without DNA amplification. Single-molecule real-time (SMRT) sequencing from Pacific Biosciences and nanopore sequencing by Oxford Nanopore Technologies are the two long-read sequencing technologies capable of detecting DNA base methylation [67,68,69]. Both methods can discriminate information between parental alleles, which is important to precisely characterize the CpG methylation pattern of the expanded CTG tract [132]. The two systems rely on very different technologies to detect CpG methylation on long reads. SMRT sequencing technology detects CpG methylation through subtle changes in polymerase kinetics throughout DNA synthesis [133]. On the other hand, nanopore sequencing technology directly detects CpG methylation via the difference produced in the electric current intensity between an unmodified and a methylated base upon passage through the pores [134,135]. Use of these two methods, PacBio and Oxford nanopore sequencing, to determine the exact CpG methylation at the DM1 locus, quantify the number of repeats, identify CNG interruptions and estimate somatic mosaicism could provide new insights into the understanding of DM1 pathogenesis and allow for the development of more personalized treatment for DM1 patients.

## Figures and Tables

**Figure 1 ijms-23-03477-f001:**
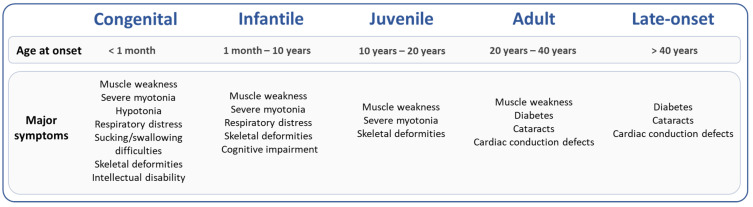
DM1 clinical classification: five forms and their main symptoms [4].

**Figure 2 ijms-23-03477-f002:**
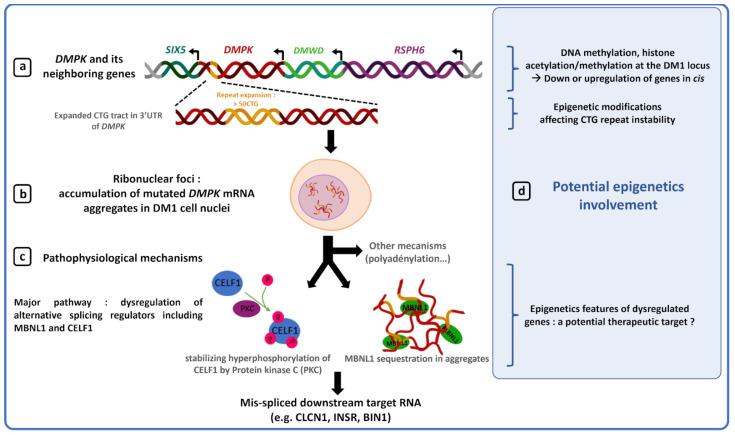
Molecular pathogenesis of DM1. (**a**) DM1 locus [49,50,51,52,53]. (**b**) Mutated mRNAs aggregation forms toxic ribonuclear foci [60]. (**c**) DM1 physio-pathological mechanisms [60,61,62,63]. (**d**) Epigenetics involvement and potential therapeutic targets in DM1 disease.

## Data Availability

Not applicable.

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
