# Peer review of "Overview of the Complex Relationship between Epigenetics Markers, CTG Repeat Instability and Symptoms in Myotonic Dystrophy Type 1"

_ijms, 2022, doi:10.3390/ijms23073477_

Round 1

Reviewer 1 Report

This review by de Pontual and Tome provides a review of the literature on epigenetics and CTG repeat instability as it pertains to myotonic dystrophy type 1. This review is fairly well written and provides a good reflection on how epigenetics and DNA instability contribute to disease onset and severity with some discussion on biomarkers and clinical relevance. Below are some suggestions to improve the review.

1) The review reads a little bit like a checklist of relevant features in DM1. More should be done to integrate the various sections better. For example, the authors could organize the review into Genetics, Epigenetics and Therapies. In this format, the authors could highlight the limitations of repeat length and interruptions as predictors of disease onset and severity and then highlight how epigenetic findings have shed light on disease. The authors can then summarize  therapies targeting epigenetic mechanisms in DM1.

2) The titles need to be more descriptive. As it stands, they simply list various topics. 

3) The use of epigenetic alterations as biomarkers seems premature in DM1 and should be toned down in the review. It only needs a brief discussion stating the potential but clear limitations that exist.

Minor edits:

Line 26 - 27: DM1 is the most common adult muscular dystrophy rather than neuromuscular disorder

Line 27: prevalence of 1 in 2100 is in New York State rather than US

Line 80: I would change "However, loss-of-function of DMPK and its flanking genes cannot explain all..." to "However, loss-of-function of DMPK and its flanking genes cannot explain the majority ..."  

Line 93-94: I would change "...show how epigenetic modifications..." to "...discuss how epigenetic modifications..."

Line 137:  (ePal) needs to be defined first time: Estimated progenitor allele length (ePAL).

Line 192: "control individuals" should be "unaffected individuals"

Line 212: What does "DM1 clinics" mean? Clinical features?

Line 264: "...an eventual role..." should be "...a potential role..."

Author Response

Author's Notes to Reviewer 1

Please find enclosed with our online submission, our revised manuscript entitled « Overview of the complex relationship between epigenetics markers, CTG repeat instability and symptoms in Myotonic Dystrophy type 1». We would like to thank the reviewer 1 for his thoughtful comments. We hope that all his concerns were adequately addressed. We added our answers in bold after each comment and modified some parts of our manuscripts (in red).

Please, see our answers below.

Best regards,

Dr Tomé

This review by de Pontual and Tome provides a review of the literature on epigenetics and CTG repeat instability as it pertains to myotonic dystrophy type 1. This review is fairly well written and provides a good reflection on how epigenetics and DNA instability contribute to disease onset and severity with some discussion on biomarkers and clinical relevance. Below are some suggestions to improve the review.

1-The review reads a little bit like a checklist of relevant features in DM1. More should be done to integrate the various sections better. For example, the authors could organize the review into Genetics, Epigenetics and Therapies. In this format, the authors could highlight the limitations of repeat length and interruptions as predictors of disease onset and severity and then highlight how epigenetic findings have shed light on disease. The authors can then summarize therapies targeting epigenetic mechanisms in DM1.

We appreciate the reviewer’s positive comment and suggestion regarding the organization of the review. Here, our review is included in the special issue of IJMS journal Epigenetic Alterations in Neuromuscular Disorders. Recently, our Italian colleagues have written a nice review entitled "Epigenetic of Myotonic Dystrophies" following the organization suggested by the reviewer (Virginia Veronica Visconti et al. 2021, IJMS). To complement our colleagues' data, we decided to organize the review differently by providing additional information. First, we focused the review on how epigenetic modifications could serve as predictive biomarkers in the DM1 phenotype (section 2). We also described how they are involved in the DM1 phenotype by changing the dynamics of repeat instability and downstream gene expression (section 3 and 4). We decided not to change the plan of the review to avoid redundancy with our Italian colleagues and to bring another approach on the requested topic. We hope that the reviewer will accept our point of view.

2) The titles need to be more descriptive. As it stands, they simply list various topics. Epigenetic data are not always consistent in DM1. The data depend on the DM1 cohort and the methods/strategies used by the authors. In addition, different information is noted in the same section. We have changed some titles where possible.

We have replaced “2.1. Specificity of congenital DM1” by “2.1 CDM1 form associated with hypermethylation of the DM1 locus and large CTG repeat expansions”

We have replaced “3.1 DM1 patient gender” by “DNA methylation at the DMPK locus may contribute to sex differences in DM1 symptoms”

We have replaced “3.2 CTG repeat instability and DM1 clinics” by “3.2 DNA methylation alters the dynamics of CTG repeat instability- Role in the DM1 phenotype?

We have replaced “3.3 CTG repeat interruptions and DM1 clinics” by “3.3 DM1 locus methylation deregulated in patients with an interrupted expanded allele and less severe symptoms”.  

3) The use of epigenetic alterations as biomarkers seems premature in DM1 and should be toned down in the review. It only needs a brief discussion stating the potential but clear limitations that exist.

We fully agree with the reviewer. More studies need to be done in DM1 to identify new epigenetic biomarkers in this disease as discussed in the section “Discussion and conclusion”. We have also discussed the limitations of the epigenetic alterations as biomarkers in DM1 in this section. See sentences below.

Line 356-358- Although promising, the use of methylation as biomarker to refine prognosis and improve treatments must be taken with caution as various external and internal factors such as the environment or the patient's age at the time of sampling are known to affect methylation levels [11,23,128-131].

Line 366-367- Analyses of epigenetic modifications are not straightforward and require homogenization of data through better characterization of the DM1 mutation, DM1 clinics and methylation status at the DM1 locus.

To complete our discussion, we added the sentence line 358-361 “In addition, epigenetic biomarkers are usually analyzed in the whole blood of patients. However, epigenetic alterations are known to be tissue-dependent. Therefore, it may be difficult to associate epigenetic markers in blood with the multisystemic DM1 symptoms and the dynamics of CTG repeat instability, which vary from tissue to tissue”

Minor edits:

Line 26 - 27: DM1 is the most common adult muscular dystrophy rather than neuromuscular disorder- We fixed it in the text.

Line 27: prevalence of 1 in 2100 is in New York State rather than US- We fixed in the text.

Line 80: I would change "However, loss-of-function of DMPK and its flanking genes cannot explain all..." to "However, loss-of-function of DMPK and its flanking genes cannot explain the majority ..."  We changed “all” by “the majority of” in the text.

Line 93-94: I would change "...show how epigenetic modifications..." to "...discuss how epigenetic modifications...". We changed “In this review, we will discuss the effects of epigenetic alterations on the clinic and genetics of DM1 and show how epigenetic modifications could serve as predictive biomarkers to improve diagnosis, prognosis, and therapies in this disease” by “In this review, we will address the effects of epigenetic alterations on the clinic and genetics of DM1 and discuss how epigenetic modifications could serve as predictive biomarkers to improve diagnosis, prognosis, and therapies in this disease”

Line 139:  (ePal) needs to be defined first time: Estimated progenitor allele length (ePAL). We fixed this point. It is quoted only once. We have not added the abbreviation

Line 193: "control individuals" should be "unaffected individuals": We fixed this point.

Line 213: What does "DM1 clinics" mean? Clinical features? We changed the title as suggested by the reviewer.

Line 266: "...an eventual role..." should be "...a potential role...": we fixed it.

Reviewer 2 Report

The article  describes in great detail the relationship between epigenetics markers, CTG repeat instability and symptoms in Myotonic Dystrophy type 1.

The topic has been carefully developed, and the authors also took care to present future perspectives for the use of knowledge.

The topic has been very professionally developed, and the authors also took care to present future perspectives of using the knowledge contained in this article.

Moreover, the authors used many literature sources, which makes their writings very rich.

The only remark is the lack of sources from which the graphics were used or based on which it was developed – FIGURE 2.

After providing this information, I think the article may be published in JIMS.

Author Response

Author's Notes to Reviewer 2

We would like to thank the reviewer 2 for his review and nice comments. We have added the literature sources (in red) we used to make the illustration in the legend to Figure 2.

Best regards,

Dr Tomé

Reviewer 3 Report

Overall, the review provides useful information on the significance of DNA Methylation and CTG repeats in DM1 disease biology. Nonetheless, more advanced research on this topic in the future will help us better understand the etiology of DM diseases.

  • Line 23: if this is a self-citation, Authors can still cite using a Reference number. I do not understand the purpose of stating Reviewed in 1, rather than saying 1.
  • Line 21-23: The sentence could be paraphrased to make it better. I wonder if the author means to say that mutation specifically in immune genes is one of the causes of neuromuscular disorders or do they mean gene mutation anywhere in the biological system?
  • Line 44: I'm curious if there is any evidence if pathogenic CTG repeat changes over the age in the same individual? 
  • I was also wondering if the epigenetic marks in DM1 are related to gender?
  • Line 40-57: Considering myself not an expert in DM1 disease biology as the authors are, I'm thinking if it exists, and could be discussed here in terms of variations in CTG repeat size and their effect on disease phenotype over the age and growth in the same individual, similarly as authors discussed the intergenerational instability. My opinion could be wrong though - in case, never mind.
  • Line 100: Do we know anything about DNA hydroxymethylation mechanism also being a plausible factor? Since aberrant hydroxymethylation is currently emerging as a prominent epigenomic biomarker in several neuropsychiatric diseases, I was curious to learn their role in DM1 disease biology if it is known.
  • When discussing potential epigenetic changes, I believe it is reasonable to discuss how hydroxymethylation associated with C/CCTG expansions could be another molecular basis of DM pathogenesis.  

Author Response

Author's Notes to Reviewer 3

Please find enclosed with our online submission, our revised manuscript entitled « Overview of the complex relationship between epigenetics markers, CTG repeat instability and symptoms in Myotonic Dystrophy type 1». We would like to thank the reviewer 3 for his thoughtful comments. We hope that all his concerns were adequately addressed. We added our answers in bold after each comment and modified some parts of our manuscripts (in red).

Please, see our answers below.

Best regards,

Dr Tomé

Overall, the review provides useful information on the significance of DNA Methylation and CTG repeats in DM1 disease biology. Nonetheless, more advanced research on this topic in the future will help us better understand the etiology of DM diseases.

Line 23: if this is a self-citation, Authors can still cite using a Reference number. I do not understand the purpose of stating Reviewed in 1, rather than saying 1. We fixed this point. We also changed 1- line 258 “[reviewed in 54]” by [54] and 2- “[for more details, see the review 99]” by “[99]”

Line 21-23: The sentence could be paraphrased to make it better. I wonder if the author means to say that mutation specifically in immune genes is one of the causes of neuromuscular disorders or do they mean gene mutation anywhere in the biological system?

Thank you to the reviewer for his comment which sparked an interesting discussion in our group. Among neuromuscular disorders, Myasthenia gravis results from autoimmune response against the neuromuscular junction and muscle tissue. Patients produce pathogenic autoantibodies. No genetic origin is known at this time. Neuromuscular diseases can be caused by a genetic mutation or by abnormalities in the immune system. We have changed “Neuromuscular disorders (NMD) include diseases affecting the peripheral nervous system and the skeletal muscles caused by either an immune system deficits or gene mutations” by “Neuromuscular disorders (NMD) include diseases affecting the peripheral nervous system and the skeletal muscles caused by either gene mutations or immune system abnormalities”. We hope this minor change makes the sentence less confusing.

Line 44: I'm curious if there is any evidence if pathogenic CTG repeat changes over the age in the same individual? 

This is a relevant question. The number of pathogenic CTG repeats increases with age in DM1 patients. Several papers have shown that the size of the CTG repeat expansion increases in blood of the same individual (Martorell et al 1995, J Med Genet; Wong et al 1995 Am. J. Hum. Genet. and Martorell et al 1998 Human Molecular Genetics). We added the literature sources on line 59 and we double-checked all the references in the manuscript. More recently, the degree of somatic mosaicism has been shown to correlate with DM1 symptom progression and age of onset as described on line 63.

We added references

I was also wondering if the epigenetic marks in DM1 are related to gender?

To my knowledge, no study has distinguished epigenetic variations between women and men in the DM1 population. For example, Breton et al. and Légaré et al. analyzed epigenetic variations in 103 and 90 male and female DM1 patients combined (F/M), respectively. In the future, this type of study could be considered for larger cohorts of DM1 patients in order to separate males from females and identify gender-associated epigenetic marks in DM1.

Line 40-57: Considering myself not an expert in DM1 disease biology as the authors are, I'm thinking if it exists, and could be discussed here in terms of variations in CTG repeat size and their effect on disease phenotype over the age and growth in the same individual, similarly as authors discussed the intergenerational instability. My opinion could be wrong though - in case, never mind.

Somatic mosaicism is an important parameter in disease progression. We have briefly described this in the review line 221-222. We added in the text line 223-224 “Furthermore, an increase in somatic mosaicism in the blood of the same individual over time is also associated with a change in the age of onset [23]” to complete this section.

Line 100: Do we know anything about DNA hydroxymethylation mechanism also being a plausible factor? Since aberrant hydroxymethylation is currently emerging as a prominent epigenomic biomarker in several neuropsychiatric diseases, I was curious to learn their role in DM1 disease biology if it is known.

To my knowledge, DNA hydroxymethylation has not been analyzed at the DM1 locus in patients. It has been shown that the distribution of 5hmC is tissue specific and not random. Analysis of DNA hydroxymethylation at the DM1 locus may provide a new understanding of the function of epigenetic modifications in the genetic and clinical variability observed in DM1 patients. Combining DNA hydroxymethylation enrichment by immunoprecipitation with long-read sequencing would give us the opportunity to evaluate this.

When discussing potential epigenetic changes, I believe it is reasonable to discuss how hydroxymethylation associated with C/CCTG expansions could be another molecular basis of DM pathogenesis.

We discussed this in the discussion and conclusion section. We have added a paragraph line 350-356 “Interestingly, DNA hydroxymethylation has not been studied in DM1 patients. Recently, aberrant DNA hydroxymethylation changes have been observed in cancers and neurological diseases such as Alzeimer’s disease suggesting an important role of the epigenomic mark hydroxymethylcytosine (5hmC) in tumorigenesis and the neuronal system (Li et al. 2021 Methods). 5hmC plays a key role in different pathways such as initiation of demethylation (Hackett et al. 2013 Science and Klug et al 2013 genome biology) and gene expression regulation (Wu et al. 2011 Genes dev and Colquitt et al. 2013 Proceedings of the National Academy of Sciences) in mammals. Analysis of DNA hydroxymethylation at the DM1 locus may provide a new insight into the function of epigenetic modifications in the dynamics of repeat instability but also in the pathogenesis of DM1”.

Line 369. We added DNA methylation and hydroxymethylation in the text.

Round 2

Reviewer 1 Report

Thank you to the authors for clarifying the accompanying review. This point addresses the organizational concern.